# Development of a Hip Joint Socket by Finite-Element-Based Analysis for Mechanical Assessment

**DOI:** 10.3390/bioengineering10020268

**Published:** 2023-02-18

**Authors:** Ana Karen González, Juvenal Rodríguez-Reséndiz, José Eli Eduardo Gonzalez-Durán, Juan Manuel Olivares Ramírez, Adyr A. Estévez-Bén

**Affiliations:** 1Engineering Faculty, Universidad Autónoma de Querétaro, Querétaro 76010, Mexico; 2Electronics Engineering Department, Instituto Tecnológico Superior del Sur de Guanajuato, Guanajuato 38980, Mexico; 3Department of Renewable Energy, Universidad Tecnológica de San Juan del Río, Querétaro 76800, Mexico; 4Chemistry Faculty, Universidad Autónoma de Querétaro, Querétaro 76010, Mexico

**Keywords:** hip disarticulation, ergonomic design, mechanical design, prosthesis, finite element analysis, joint biomechanics, computational-modeling-based design

## Abstract

This article evaluates a hip joint socket design by finite element method (FEM). The study was based on the needs and characteristics of a patient with an oncological amputation; however, the solution and the presented method may be generalized for patients with similar conditions. The research aimed to solve a generalized problem, taking a typical case from the study area as a reference. Data were collected on the use of the current improving prosthesis—specifically in interaction with its socket—to obtain information on the new approach design: this step constituted the work’s starting point, where the problems to be solved in conventional designs were revealed. Currently, the development of this type of support does not consider the functionality and comfort of the patient. Research has reported that 58% of patients with sockets have rejected their use, because they do not fit comfortably and functionally; therefore, patients’ low acceptance or rejection of the use of the prosthesis socket has been documented. In this study, different designs were evaluated, based on the FEM as scientific support for the results obtained, for the development of a new ergonomic fit with a 60% increase in patient compliance, that had correct gait performance when correcting postures, improved fit–user interaction, and that presented an esthetic fit that met the usability factor. The validation of the results was carried out through the physical construction of the prototype. The research showed how the finite element method improved the design, analyzing the structural behavioral, and that it could reduce cost and time instead of generating several prototypes.

## 1. Introduction

The main causes of lower limb amputations are malignancy and vascular disease [1]. According to Mexican Academy of Surgery (CRIMAL) data, approximately 75 amputations are performed in the country each day, representing 27,735 patients per year. The National Institute of Statistics, Geography and Informatics (INEGI, by its acronym in Spanish) showed in its latest data that there are a total of 2,437,397 people with walking or moving disabilities in Mexico [2]. In most cases, the amputation of a person’s lower limbs also affects the junction with their new prosthesis [3].

Affected patients with a disability generally face difficulty integrating into their daily tasks, from a social perspective [4]. In [5], it is mentioned that, after completing the rehabilitation period, patients experience falls at least once a year, with a 33% probability of multiple falls. When the wound has healed properly, and the necessary rehabilitation is carried out so that the patient can bear the weight of it, the use of a foot prosthesis is generally recommended [6]; therefore, it is necessary to implant orthoses and prostheses, which help to recover the functions performed by the amputated limb [7].

The development of supports must consider the functionality and comfort of the patient, as they are critical aspects of its acceptance. In [8,9], the essential aspects that determine the level of patient satisfaction are summarized, from which it can be concluded that it is essential to take into account aspects such as the length and shape of the stump, muscle strength, areas of pain and the appropriate length of the prosthesis, to avoid adverse effects on the patient in the adaptation of the device, and to achieve a design that fits correctly: in this sense, although the materials currently used are generally lightweight, a socket has not been developed to comfortably contain this constantly changing part of the body [10].

In patients with hip disarticulation, the remaining end is known as the stump. The stump is a limb with different shapes, depending on the amputation performed, so the socket design must be a process that considers the physical needs of each patient [11]. A poorly fitting socket generally leads to rejection of the prosthesis, due to irritation, pain, skin ulcers and other medical conditions that can be aggravated, depending on the situation [12,13]. Scientific development has made it possible to create mechanisms capable of reducing rejection rates; however, as it is not a personalized technique for each patient, there is still a high rejection rate [14,15]. Some aspects considered essential to the investigation are detailed below.

Today, different types of socket are used, depending on the level of amputation [16,17,18,19]. A plug has been found—patent WO2016157983A1—that allows the user to sit in a better way [20]: it is used if the patient has had both legs amputated. Patent US11051954 [21], which was developed in 2007, is one of the most recognized accessories: it eliminates some pressure points. However, the aforementioned patents constitute the final development of several stages of research: for example, Ref. [22] evaluated the design of a variable impedance prosthetic socket (VIPr) for a transtibial amputee, using computer-aided design and manufacturing (CAD/CAM) processes, while Ref. [23] presented possible shape optimization methods that could benefit the prosthetist and limb user, by combining state-of-the-art prosthetic mechanical design, surrogate modeling and evolutionary computing. In the literature, works such as [24,25] could help to analyze the kinematics of the joint in the amputated leg. Other works support the current investigation, and constitute an obligatory reference by which to give scientific support to the proposed development: these are summarized in Table 1.

In this research, the critical elements for developing a socket were identified: weight distribution; empathic design; ergonomics and anthropometry—points all taken into account in the design. The design was analyzed under mechanical design criteria, through finite element analysis (FEA), due its complex geometry [35]. FEA was used by [36] to investigate anterior cruciate ligament (ACL) stress distribution under different loading conditions, and by [37], to evaluate biomechanical characteristics, using FEA to confirm the new abutment system’s clinical applicability: this concluded with the validation of the new adjustment, which was carried out through a perception survey focused on the author’s theory of comfort [38], where comfort was evaluated on the basis of the different activities that the patient performed with both sockets, before and after the improved design. According to the reports made to the patient, an increase of more than 60% was observed in the comfort level. Another interesting work, by [39], developed a new helmet, using FEM, which showed improvements in impact energy attenuation, as well as in kinematic and biometric injury risk reduction. Ref. [40] proposed FEA as an adjunct to preoperative imaging, to assist patient selection and procedure planning, as well as to help in the detection and prevention of transcatheter heart valve replacement complications. Some other works, such as [41], provide reference values for the usual viscoelastic properties, which would allow more accurate numerical simulation by means of FEM, for example.

The manuscript is structured as follows. In Section 2, the methodology or scientific procedure used in the research to develop the new ergonomic fit is presented. The main results obtained are shown, detailing various aspects, such as data collection, analysis, socket design by FEM and socket manufacture. Section 3 presents the different models, analyzed using the FEM through simulation technique. Section 4 presents the validation of the socket development, the study of the results obtained and its discussion. The validation stage is essential to verifying the increase in the comfort of the user sensation. Finally, work conclusions are presented.

## 2. Method

Male subjects with an age range of 20 to 38 years were studied. Different studies have determined that this population is the most involved in amputations [42]; in addition, these people are in a weight range less than or equal to 65 kg, with regard to hip disarticulation and active prosthesis.

The user profile was prepared by considering the cause of the amputation, the moment it occurred, the fitting stage and the adaptability of the prosthesis. The following evaluation objects were considered: (a) the Pohjolainen classification [43], which is based on seven scales, and indicates whether the patient can walk independently or needs some other technical help; (b) the Houghton Instrument [44], which determines if the rehabilitation is satisfactory, and consists of four questions, which together have a maximum score of 12 points, where 9 points is considered a satisfactory rehabilitation; (c) the motor capacity index [45], which determines the functionality of the prosthesis. The above analysis was applied to the patients, to ascertain possible points for improvement in the design.

After analyzing the sample, a 29-year-old male patient was selected, with a body mass of 65 kg and an oncological amputation, taking into account that he met the average profile of the sample; however, the design can be generalized for patients with different physiques: this aspect— the adaptability of the procedure to patients with different characteristics—is one of the main contributions to the research study field. The patient’s amputation was three years old, including one year of protection where the adaptability to the prosthesis was low.

In the advanced stages of the project, tests were carried out by the Rapid Upper Limb Assessment (RULA) used by [46], and by the Rapid Entire Body Assessment (REBA) [47] methods, on both types of sockets (conventional and ergonomic), with the aim being to validate the ergonomic socket developed, in such a way as to measure the improvement against conventional sockets. The tests were carried out by walking on flat ground for 6 min, and approximately 200 photos were taken, selecting the postures that were repeated most frequently. After analyzing the images by the RULA method—by which the angles that the user acquired with the conventional socket and with the ergonomic socket could be defined—a selection was made with the posture that presented the highest risk.

In the case of REBA with a conventional socket, it was found that the postures acquired by the user on the right side (the side of use of the prosthesis) and on the left side, presented a medium level of risk, when obtaining a score of 4. REBA analyzes two segments of the body: group A, which is made up of the trunk, neck and leg; and group B, which refers to the arm, forearm, and wrist.

In the case of REBA with an ergonomic fit, the score obtained was 2: this was a low-risk level, so action was not necessary. With this, the new fit improved the postures and angles of both the left and right sides, when developing the gait. By analyzing RULA with a conventional socket and ergonomic socket, it was possible to define the angles that the user acquired with both sockets. Walking and sitting tests carried out were. From Figure 1, it can be seen that the user had an incorrect posture using the conventional socket, as the angle was $234^{\circ }$–$126^{\circ }$, while, with the ergonomic socket, the user’s posture improved, with the user maintaining a lower-risk posture, according to RULA and REBA, when walking, so that the angle was reduced to $215^{\circ }$–$145^{\circ }$. This meant that the conventional socket did not comply with the ergonomic parameters.

Figure 2 shows the analysis of the knee flexion angle of the prosthesis: with the conventional socket, a flex of $207^{\circ }$ was obtained; with the ergonomic socket, the flex increased only to $208^{\circ }$; therefore, it can be concluded that it is necessary to improve the fulcrum of the pelvis when walking, as this is what provides the force to carry out the walk.

### 2.1. Data Collection

The evaluation objects and their results, for the development of the prototype, are in the Appendix Section. The user, according to the Pohjolainen classification shown in Table A1, was class II, which means that he walked independently at home with his prosthesis, but that he required technical help when outdoors, using a pair of crutches. The Houghton Instrument illustrated in Table A2 consists of four questions, which together have a maximum score of 12 points, with 9 points being considered a satisfactory rehabilitation. The resulting score was 7, which meant that the patient had not successfully achieved his rehabilitation. This instrument indicated some of the critical factors regarding walking, percentage of use of the prosthesis, and terrain types over which the user could move.

On the other hand, through the Motor Capacity Index presented in Table A3, 12 activities were evaluated, with the possible answers being: “NO”—with a value of 1 point; “YES, with the help of someone”—with a value of 2 points; “YES, supported by an object”—with a value of 3 points; and “YES”—with a value of 4 points. Subsequently, the values were added; the maximum value was 37, when all the affirmations were answered positively. The patient needed some object to perform most activities, which was why the prosthesis did not adequately fulfill the function of replacing the amputated limb.

The prosthetic satisfaction questionnaire (SAT-PRO), in Table A4, is a tool that measures the satisfaction of lower limb amputees, in relation to their prosthesis. The questionnaire is designed for use after prosthetic training. The questionnaire revealed, through the answers, that the user was not very satisfied with his fit, as he had suffered some injuries. The analysis of the data obtained showed that the user was most affected in regard to the factor of tranquility when making use of his socket, as the four activities—walking, sitting, standing and going up and down stairs—had a higher score of nonconformity.

### 2.2. Data Analysis

The data obtained showed that the user was affected, in terms of tranquility, when using his plug. Going up and down stairs had caused injuries to the user, in the areas of the stump and the abdomen. The patient had to make a greater effort when performing this activity, due to the pain it caused: this effort generated inappropriate postures, in order to prevent the lace from rubbing in the affected areas. The symptoms that were described by the selected person were also reported by the sample of patients analyzed. The total comfort percentage per activity appears in Table 2.

In regard to the four activities (walking, sitting, standing, going up and down stairs), they had a higher score of non-conformity. In addition, in the perception questionnaire, it was evidenced that the stump was subjected to stress while performing activities with the socket. On this basis, there was a need to create an ergonomic and comfortable environment, so that the stump did not come into direct contact with the semi-rigid material. Finally, the user expressed the need to modify the comfort of his current fit.

Going up and down stairs were the activities with the lowest percentage of comfort, for the reasons already mentioned (8%), followed by standing or sitting (17%) and walking (25%). The overall comfort percentage of the conventional socket was 6.75%, being deficient, and showing injuries and discomfort in the user.

Visualization of the quality function (QFD) was carried out, and the results are presented in Table 3, knowing the deficiencies of the conventional adjustment. The most crucial aspect for the patient was autonomy, which entailed a prosthesis that allowed him to feel safe while wearing it. In addition, the user required stability, i.e., that the prosthesis conformed anatomically to the user’s shape. When walking or sitting, the prosthesis should not be prone to movements that could cause the patient to fall. The esthetic characteristics of the socket needed to be taken into account. Finally, the patient described the need for independence. Thus, all the information necessary for the development of the ergonomic fit was captured.

The results that were considered in the development of the socket are presented in Figure 3a. For the optimization of weight and shape, it was advisable to eliminate some surfaces. The methodology used to decide whether to delete a surface is found in Figure 3b. Finally, when changing the designs, some surfaces became smaller, and the loads were concentrated, for which mechanical reinforcement was necessary, to support the loads before failure: this entailed considering a safety factor lower than 1, for which the methodology is described in Figure 3c.

In Figure 3 above, B is directional deformation (X-axis), directional deformation (Y-axis) and directional deformation (Z-axis), taking into account the change in the size or shape of a body due to the internal stresses produced by one or more forces applied to it, with total deformation being the square root of the sum of the squares of the deformation on each axis. The structural error shows the maximum error occurrence on the elements of the mesh: we could refine the mesh structure according to these error results. A safety factor was used, such as yield stress divided by the maximum stress result from FEM. Life shows the available life for the given fatigue analysis with a load of constant amplitude: this represents the number of cycles until the part failed due to fatigue. For the fatigue safety factor, values less than 1 indicate failure before the design life was reached. Equivalent stress—the stress due to loads applied on the model, and strain energy—was the energy requirement to store the total strain on the whole body. C was the potential surface to eliminate. E was the insert support.

### 2.3. Socket Design

The socket had to support the patient’s weight: this was achieved by weight distribution and, where necessary, by reinforcement in critical areas. Another element identified was comfort, which was achieved by the correct choice of materials, especially in the padding area. The shape of the socket had to conform to the morphology of the user. The esthetic was divided into three factors: color, design and size. It was necessary to select the correct color, a non-intrusive design and a size that allowed the plug to function without compromising the user’s choice of clothing. A representative number of preliminary designs were carried out, based on the requirements mentioned above. Section 3 and following describe the evolution and analysis of the three most important designs, starting with the conventional socket, using the same boundary conditions for structural evaluation through the FEM, and using commercial software according to the methodology described in Figure 3b,c.

Critical support and suspension zones were defined using data obtained from the sensors system that was implemented in the conventional socket, when under use by the user, obtaining the critical forces generated by the four activities (walking, sitting, standing, going up and down stairs).

The volumes were meshed and, finally, a mechanical analysis of the system was performed. The weight of the patient determined the minimum load, and with the distribution of the weight, the design was optimized by means of static structural analysis, directional deformation (X, Y and Z Axis), total deformation, structural error, safety factor, life, equivalent stress and strain energy. For optimizing the load distribution, the following materials were considered: 30% wt glass-fiber-reinforced polypropylene, and aluminum (Al-6061). The properties of materials such as Young’s Modulus (*E*), Poisson’s Ratio (υ), density (ρ) and Tensile Yield Strength ($\sigma_{y}$) are shown in Table 4, together with the simulation parameters used as boundary conditions for each model evaluated, such as loads and supports, where they are a simplification of the connection for the artificial limb and the socket. For fatigue analysis, the mean stress theory was used by Soderberg, because it used yield stress instead of ultimate stress with a fully reversed load.

The sensibility analysis for the mesh was realized, and the mesh described presented a minimal strain variation. Figure 4a displays the mesh for Model 1, which corresponded to the user’s conventional socket. A shell element with a thickness of 5 mm was used, with a delicate refinement in the center. A face sizing of 0.5 mm was applied, obtaining 293,865 nodes and 291,972 elements.

The mesh generated for Model 2 is shown in Figure 4b—the result of an intermediate improvement of the conventional socket. As for meshing options, the following were selected: a 1.5 mm element size, the hex-dominant method, a quadratic element order, a quad/tri-type free face mesh, and fine refinement in the center, obtaining 969,298 nodes and 254,482 elements.

Figure 4c illustrates the mesh of Model 3, the result of the final prototype of the ergonomic socket. As for meshing options, the following were selected: a 1.5 mm element size, the hex-dominant method, a quadratic element order, a quad/tri-type free face mesh, a 0.5 mm body sizing applied to the aluminum reinforcement part, and fine refinement in the center, obtaining 3,060,273 nodes and 829,810 elements.

## 3. Analysis by FEM

The FEM is one of the most advanced simulation techniques in solid mechanics, and is used for the optimization of orthopedic prostheses. The prosthetic design included distribution of the weight obtained by loads from a FlexiForce Pressure Sensor (25 lb–force 3% for linearity and accuracy) placed on the iliac crests, the front part of the abdomen and the lower part, where the stump was recharged, to measure the position and force exerted by the person against the socket used in simulation. These loads are described for Model 1, shown in Figure 5a. The location of the loads was as follows: Strength B (−543.72 N), the surface where the stump of the patient was supported, considering the y-axis of the coordinates; Strength C (−127.34 N), the lateral load surface for the journal, considering the x-axis of the coordinates; Strength D (−205.68 N), the surface where the hip of the patient was supported, referring to the x-axis of the coordinates.

After meshing, the boundary conditions depicted in Figure 5a were applied and solved by the FEM. The results for Model 1 are described below. Figure 5b shows the directional deformation, considering the x-axis, where a 0.42 mm deformation is shown in the region that supported the person’s back; however, the most significant deformation occurred on the lateral side of the hip, with a value of −1.09 mm. The deformation related to the y-axis, shown in Figure 5c, presented an amount of −1.71 mm at the base of the support, exceeding the deformation value in X. The most significant load was on the y-axis; the simulation results were compatible with the real behavior of the system. Directional deformation in the z-axis occurred in the back 1.25 mm and the abdominal area −0.78 mm. It should be noted that the abdominal height support was a surface where deformations interacted, as shown in Figure 5e. The simulation of directional deformations by coordinate axis (Figure 5b–d) was important, as it enabled the visualizing of the behavior of the system and its reactions in a better way than simulating the total deformation in Figure 5e. Cutting the surfaces where there were lower amounts of total deformation would have subtracted the surfaces with the most significant deformation, Figure 5f, which should prevail in the system.

Monitoring by simulation enabled detection of the surfaces where potential structural damage could occur. Figure 5g shows that the geometry subjected to the established forces did not cause structural damage, due to the uniformity of energy absorption, with a value of $5.47\;\times \;10^{-13}$ mJ; however, in regard to the safety factor, Figure 5h shows some red surfaces, where the safety factor was 0.36, indicating that it may have had structural failures. In these surfaces, the required mechanical capacity exceeded the maximum capacity of the material. An area of opportunity was presented for improving the system design. With regard to static loads (Figure 5i), the system could support $1\;\times \;10^{6}$ times the static load. It was convenient to subject the model to dynamic loads, and values lower than 1 continued to prevail, with regard to the safety factor, as shown in Figure 5j. Figure 5k exhibited the result of the equivalent stress (Von Mises), in which it can be seen that the maximum value was 46 MPa: as the material could withstand 33 MPa, a stress fracture was possible. Figure 5l presents the deformation energy, which was an increase in the internal energy in the system, as a result of the applied loads, with $2.25\;\times \;10^{-12}$ mJ being obtained on most of the surface.

In Figure 6a, the boundary conditions for Model 2 are shown. The location of the forces corresponds to the same position where the sensors were placed in the first model. Additional cylindrical support was added to the hole, where a screw was to be placed, to join the prosthesis. Figure 6b–d show the displacements generated due to the boundary conditions in the x, y and z axes, respectively. The z-axis displacements were proportional, with 13 mm on average for the positive and negative direction, resulting in 23% with regard to the highest value, and for the x-axis, it represented 22%. The displacements on the x-axis were the smallest of the three directions; however, the y-axis displacements were the largest, and had to be decreased, because they were in the load direction, and could be uncomfortable for the user. Figure 6e and/or Figure 6f show that the total displacement was more significant in the red area, which was part of the waist support, than in the green area at the socket base. In Figure 6g, it can be seen that the areas with displacements greater than 5.6 mm required reduction to a minimum, to achieve greater comfort for the user.

The Von Mises stresses shown in Figure 6h show that the most critical zone was the seat and screw clamping, which required actions to decrease the stress values in this zone. Figure 6i illustrates the areas that required some reinforcement to reduce Von Mises stresses, and to prevent the socket from failing. From this figure, it can be concluded that the reinforcement should be located at the socket base. Figure 6j shows that a transition from high to low energy did not occur in adjacent elements; therefore, a refinement in the meshing was not necessary, which proved the sensibility analysis of the mesh.

In the areas where the material could not possibly resist, the safety factor of the model was calculated. According to Figure 6k,l, it was necessary to reinforce the socket base, as that was where the safety factor was less than 1. As the socket would suffer from alternating stresses when walking, a fatigue analysis was necessary, to assess the number of cycles the socket could withstand. Figure 6m shows that the safety factor was even lower than in the static part, and that the areas with a value of less than 1 spread to practically the entire base. The red areas in Figure 6n represent the number of cycles that the socket could withstand, which was very low; however, the other areas show that they exceeded 1 million cycles: therefore, the damage or probable failure zones due to cyclical stresses would be in the red areas, as presented in Figure 6o.

In Figure 7a, the boundary conditions for Model 3 are shown. The location of the forces corresponds to the same position where the sensors were placed in the first model. Additionally, as seen in Figure 7a, cylindrical support for the hole would be used to secure the aluminum bracket to the socket base, where a screw would be placed to attach the prosthesis or artificial limb to the aluminum support; a condition of fixed support was applied, which restricted displacements on the axes. Figure 7b–d shows the deformation generated due to the boundary conditions in the x, y and z axes, respectively, where a notable decrease can be seen, from the maximum of 50 mm obtained in Model 2 to only 3.8 mm for Model 3. The z-axis deformations were greater in the positive direction, at 3.34 mm, very similar to the highest value on the y-axis, at 3.71 mm in the negative direction. The deformations on the x-axis were the smallest in both directions; however, the deformations in the negative direction of the y-axis were the largest, which is to be expected, because it was the direction that would carry most of the weight of the user. Figure 7e and/or Figure 7f show that the total deformation was greater in the red zone at the socket base. Figure 7g shows the areas with deformation less than 0.4 mm.

In Figure 7h, it can be seen that a stress concentration zone on the superior hole was due to geometry conditions in the model, possible positions and selected diameters in the diameter for the support screws. Figure 7i shows that the transition from high to low energy did not occur in adjacent elements; therefore, the refinement in the meshing was sufficient. In order to figure out the areas where the material could present failures, the model’s safety factor was calculated. According to Figure 7j–l, a dangerous zone was the lower hole from top to bottom, because the safety factor was less than 1.

As the socket would suffer from alternating stresses when walking, a fatigue analysis was necessary, to assess the number of cycles the socket could withstand. From Figure 7m, it can be seen that the safety factor was even lower than in the static part, and that the areas with a value less than 1 propagated beyond the lower hole, as shown in Figure 7n. In Figure 7o, it can be seen that the entire socket exceeded 1 million cycles, except for the lower hole: to improve this situation, it was decided to optimize the position and dimension of the holes in the aluminum base, the results of which are shown in Section 3.1.

### 3.1. Stress Concentration Treatment

In Model 3, the two holes generated a stress concentrator. This section shows the safety factor result when evaluating Model 3 under the boundary conditions described above. The safety factor result was used because it was a value that took into account the stresses allowed by design, regarding the maximum stresses obtained in the object of study under certain load conditions—in this case, Model 3—to achieve a safety factor of at least 1, which indicated that our maximum stress obtained under user loads did not exceed the material design stress.

Figure 8a shows that the dimension of the hole was changed below 8 mm: as a result, the safety factor was 0.49. It was decided to reduce the hole to 7 mm, as shown in Figure 8b, which increased the safety factor a little, but it was still not enough. In Figure 8c with a 5 mm hole, the safety factor was considerably away from 1. In Figure 8d, we can see how the hole was displaced towards the top: a safety factor of 0.45 was obtained. In the next attempt, the hole was moved to the flat surface of the aluminum support, as shown in Figure 8e, and the safety factor did not improve, reaching only 0.39; additionally, the hole was rounded and, as seen in Figure 8f, it worsened, reaching a value of 0.23.

The hole was displaced, as displayed in Figure 8g, and the result was similar to that obtained in the position of Figure 8f, with an amount of 0.26. Another analysis was performed by placing the holes aligned horizontally, as seen in Figure 8h, achieving 0.45—still far from the target. The number of holes was increased, in order to distribute the magnitude of the stresses, and to improve the safety factor; however, as shown in Figure 8i, the result of 0.45 was identical when the holes were aligned horizontally. The two horizontal holes were moved to the flat part of the aluminum support, as shown in Figure 8j; however, the result was the same as in Figure 8h. Of all the possible configurations, and doing a force analysis, the most significant load was vertical: when the holes were aligned in this way, the maximum effort decreased, and it was possible to achieve at least one safety factor of 1, as the Figure 8k shows. Due to the mechanical assessment, a final design proposed for development of the prototype was obtained, the validation of which is shown in Section 4.

## 4. Prototype Manufacture and Discussion

Among the materials proposed, as was mentioned above, for the design of the socket, the following materials were used to manufacture the prototype: polypropylene; aluminum; and textile neoprene. Detailed information on their use is provided below.

Polypropylene was used due to its high degree of malleability and its low density. It was used in the basket, and allowed to be semi-rigid, so that the patient could have a greater degree of freedom in the activities he performed daily. Polyethylene is a semi-rigid material, and was allowed to help in holding the basket; furthermore, due to its high mechanical resistance and lightness, the aluminum support allowed the reinforcement of critical areas throughout the socket, according to the results of simulations by FEM. The textile neoprene was used because it maintains flexibility, which is why it is used in orthopedic devices to adjust to the object or person requiring protection, and to help to reduce the contact pressure between limb and basket; also, it is resistant to degradation and damage from bending and twisting.

The process carried out to transform the raw material consisted of heating a thermoplastic sheet (polypropylene), and adapting it to the shape of a mold through the action of vacuum pressure, by means of a counter-mold. For the cast, the anatomy of the user was used. In this way, the exact shape of the stump and the affected limbs were obtained. At the end of the mold, the design was drawn, based on a previous prototype made of flexible plastic, and according to the user (the height of the hip, the areas of the iliac crest, the buttocks and the genitals). Figure 9 shows the result of the manufacture of the final prototype; Figure 9a is called the ergonomic socket, Figure 9b shows the socket tried out by the user and Figure 9c is the conventional socket.

The questionnaire was used again, to validate the prototype, and to quantify the increase in comfort in the new design. The user wore the socket for 10 days, performing his daily activities. At the end of this time, when the survey was applied, the most disagreeable activity continued to be going up and down stairs; however, it was less so, compared to the conventional socket. Regarding the transcendence, it was found that the user still did not feel fully independent, so much work must be done to improve this aspect. As can be seen from Table 5, the overall comfort of the ergonomic fit increased considerably, from 17% to 77%. The activity with the highest percentage of comfort was walking, which is essential, as it is how the person can move from one socket to another. Increased relief, reassurance and importance factors directly result in better user ambulation, avoiding injury, pain and worry.

The validation stage culminated in the Validation for User Experience (VEU) method. The manual was used to rate the user experience, with respect to products with which it interacted. The procedure consisted of different sections: for the purposes of the project, only those of usability, interaction and esthetics were used. The consistency ranges were as shown in Table 6.

In the usability section, with the conventional fit, a score of 3.5 was obtained, which was acceptable; however, there were deficiencies in the handling of the product. The user emphasized that the conventional socket was quite intrusive, and that when it was used, it was quite uncomfortable. Contrary to the conventional socket, the ergonomic socket facilitated handling, due to its mobile section on the lateral side: it adapted to the morphology of the user, so that it was quite intuitive to socket it on the stump. The total score obtained was 4.75, which showed improvements in usability.

In the interaction section, the physical reaction and pain items were alarming, as they received a low weighting. The conventional socket caused discomfort in the stump and part of the abdomen, when performing daily activities. The interaction with the conventional fit was questionable, as it received a rating of 3. Regarding the interaction with the ergonomic fit, improvement could be observed in all aspects, except in the item that addressed the development of the interaction, and it was evaluated with the same rating as the conventional socket. The overall rating was 4.4, which was in the good range. When interacting with the socket, the user did not perceive pain, and was able to perform better in his daily activities.

The last section evaluated was esthetics. The item with the lowest weighting was the one where the user perceived that the conventional socket was not pleasing to the eye. The patient was dissatisfied with the color, shape and texture of the area padding. According to the manual, the esthetics section received a low score, so it was an area with many opportunities. Ergonomic fit, meanwhile, received a reasonably favorable score, of 4.5. This indicates that the expectations of the user were met; however, improvements must be made at all times, to be able to compete in the market.

## 5. Conclusions

An improved socket for hip joints was developed, taking into account data about conventional sockets used, a user questionnaire, FEM, manufacturing the prototype to evaluate it and, finally, validating better performance than that of conventional sockets.

FEM enabled us to analyze several virtual prototypes, to achieve the best proposal for the user to evaluate. The design took into account two important parameters: displacement and the safety factor. The results showed an ergonomic, strong and durable basket, according to charge cycles, as well as the best position of the aluminum support holes to help in holding the basket: it was validated by the user wearing the socket for 10 days.

Identification of the user’s needs was essential to defining a design profile with which tangible and adequate solutions could be generated for the dis-articulated hip. Experimental measurement of the highest pressure points on the stump, using a conventional socket sensor system and FEA, allowed the development of an ergonomically functional socket, with fewer prototypes and shorter development time.

From the field research, it was found that the materials used in the area of padding the socket were not adequate, so a selection of materials was made that improved interaction, and allowed the user to develop his daily activities, without facing the problems generated by a conventional socket. The ergonomic fit was validated using different tools, obtaining increased comfort, correct gait performance (by correcting posture), improved fit–user interaction and designing an esthetic fit designed to meet the usability factor. The issues most valued by the user, under VEU, were the absence of pain, interaction and esthetics, and walking, sitting down, moving up or down and getting up were aspects improved by the ergonomic socket.

## Figures and Tables

**Figure 1 bioengineering-10-00268-f001:**
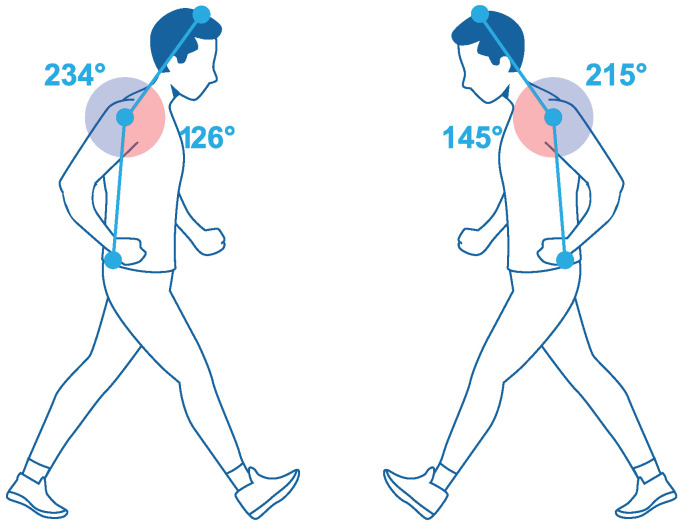
Analysis of the positions by the RULA method: (**left**) conventional socket, (**right**) ergonomic socket.

**Figure 2 bioengineering-10-00268-f002:**
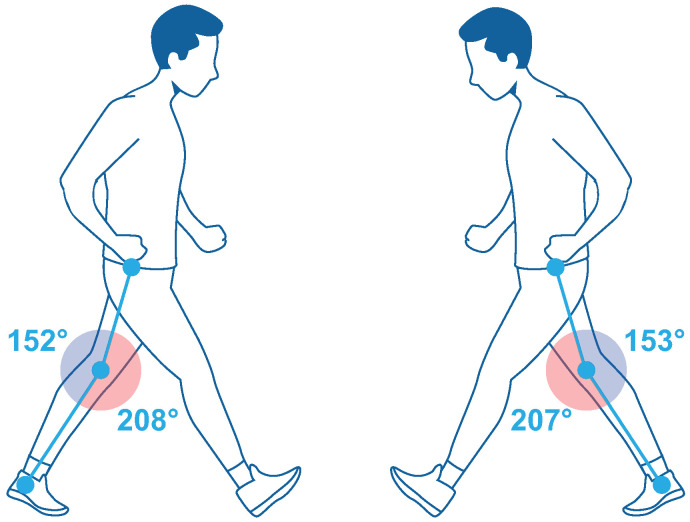
Analysis of flexion angles in the knee: (**left**) conventional socket; (**right**) ergonomic socket.

**Figure 3 bioengineering-10-00268-f003:**
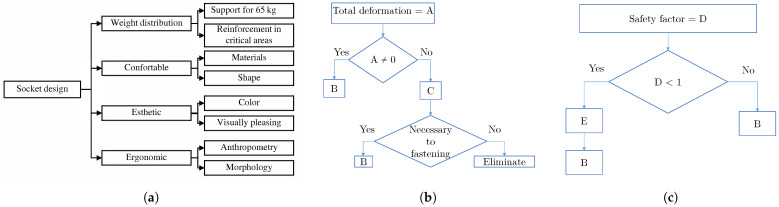
Features for socket design: (**a**) elements to consider for socket design; (**b**) methodology for surface treatment; and (**c**) methodology for insert support.

**Figure 4 bioengineering-10-00268-f004:**
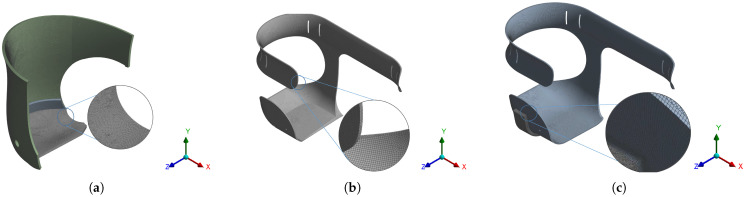
Meshing of the models presented in this work: (**a**) Model 1; (**b**) Model 2; (**c**) Model 3.

**Figure 5 bioengineering-10-00268-f005:**
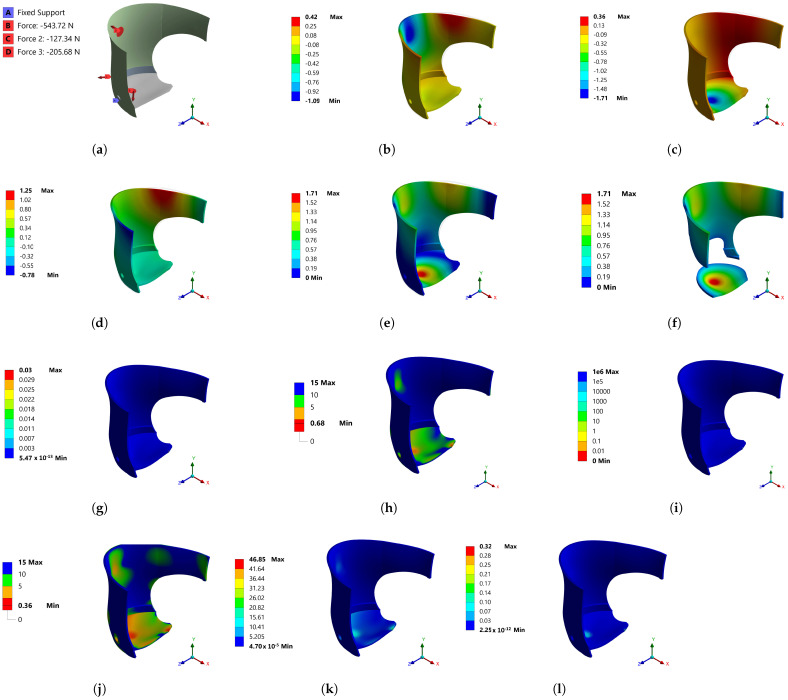
Mechanical behavior results for Model 1: (**a**) Boundary conditions; (**b**) directional deformation (mm) (x-axis); (**c**) directional deformation (mm) (y-axis); (**d**) directional deformation (mm) (z-axis); (**e**) total deformation (mm); (**f**) total deformation (mm), surfaces with greater deformation; (**g**) structural error (mJ); (**h**) safety factor; (**i**) Life; (**j**) safety factor; (**k**) equivalent stress (Mpa); (**l**) strain energy (mJ).

**Figure 6 bioengineering-10-00268-f006:**
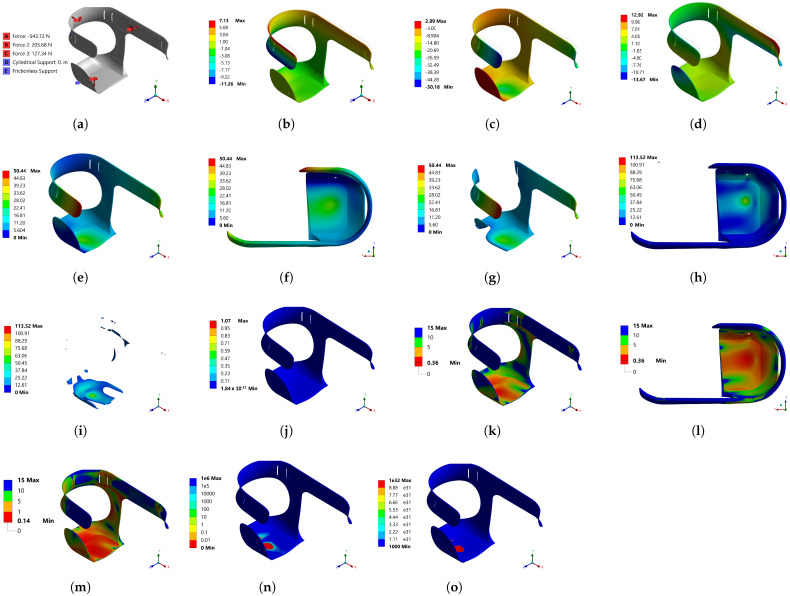
Mechanical behavior results for Model 2: (**a**) loads and supports; (**b**) directional deformation (mm) (x-axis); (**c**) directional deformation (mm) (y-axis); (**d**) directional deformation (mm) (z-axis).; (**e**) total deformation (mm); (**f**) total deformation (mm), top view; (**g**) total deformation (mm), surfaces with greater deformation; (**h**) Von Mises equivalent stress (MPa) top view; (**i**) Von Mises equivalent stress (MPa), zones with stress greater than 50 MPa; (**j**) structural error (mJ); (**k**) safety factor, static load; (**l**) safety factor, static load top view; (**m**) fatigue, safety factor; (**n**) fatigue, life; (**o**) fatigue, damage.

**Figure 7 bioengineering-10-00268-f007:**
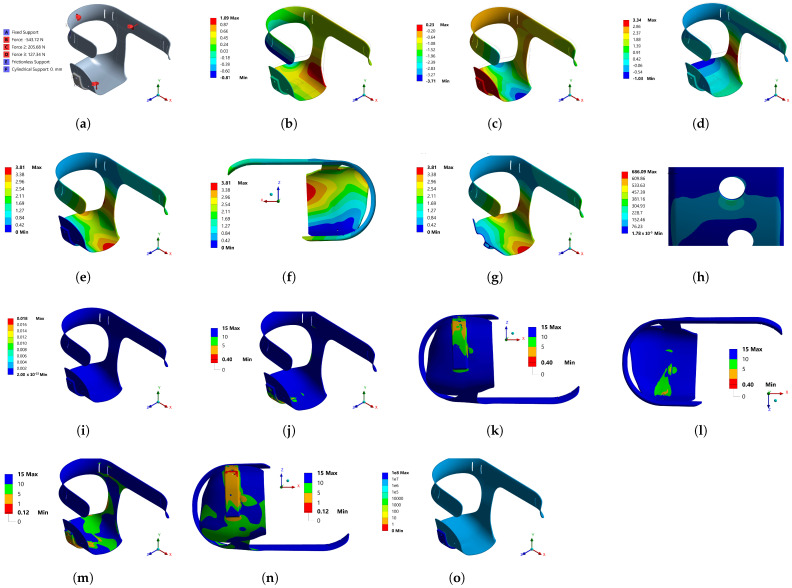
Mechanical behavior results for Model 3: (**a**) loads and supports; (**b**) directional deformation (mm) (x-axis); (**c**) directional deformation (mm) (y-axis); (**d**) directional deformation (mm) (z-axis); (**e**) total deformation (mm); (**f**) total deformation (mm), top view; (**g**) total deformation (mm); surfaces with greater deformation; (**h**) Von Mises equivalent stress (MPa), stress concentration; (**i**) structural error (mJ); (**j**) safety factor, static load; (**k**) safety factor, static load; (**l**) safety factor; static load; (**m**) fatigue, safety factor; (**n**) fatigue, safety factor bottom view; (**o**) fatigue, life.

**Figure 8 bioengineering-10-00268-f008:**
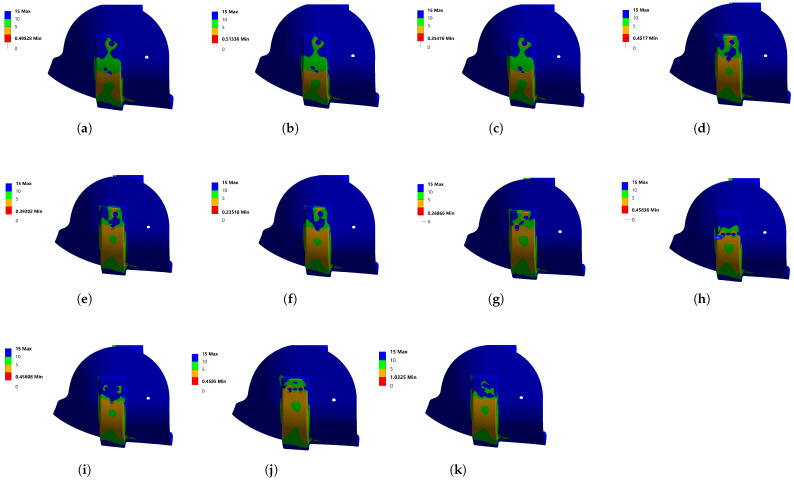
Safety factor under different scenarios of holes for Model 3: (**a**) safety factor—8 mm hole; (**b**) safety factor—7 mm hole; (**c**) safety factor—5 mm hole; (**d**) safety factor—8 mm hole, displaced; (**e**) safety factor—8 mm hole, flat surface; (**f**) safety factor—8 mm hole, flat surface and round; (**g**) safety factor—8 mm opposite side; (**h**) safety factor—8 mm horizontal holes; (**i**) safety factor—8 mm three holes; (**j**) safety factor—8 mm two superior horizontal holes; (**k**) safety factor—8 mm two vertical holes.

**Figure 9 bioengineering-10-00268-f009:**
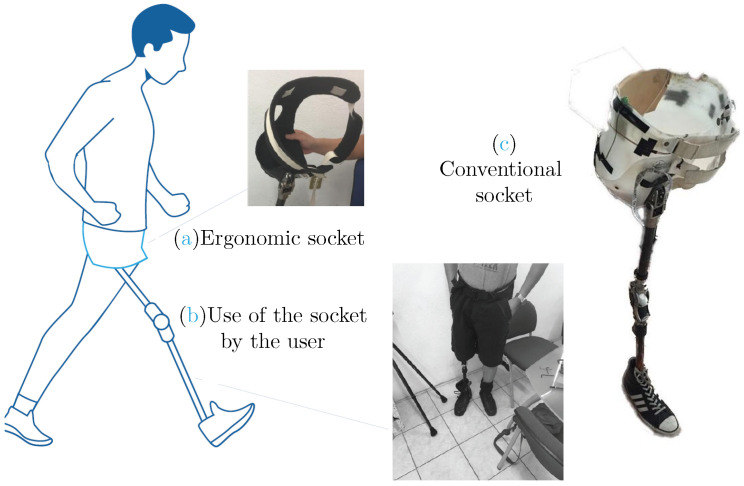
Final prototype and conventional socket.

**Table 1 bioengineering-10-00268-t001:** Use of the FEM in investigations of the area.

Reference/Year	Contribution to the Field of Study
[26]	Friction-induced instability in the hip joint system was analyzed, using the finite element method.
[27]	In the study, a finite element approach, with contact transformation, was proposed, which required less computational effort.
[28]	The authors performed a numerical simulation that was capable of predicting variations in vascular geometry.
[29]	This systematic literature review investigated the state of the art in residual limb finite element analyses published since 2000.
[30]	Finite element 3D models from an interface perfectly adapted to large conical misalignments, and a wear algorithm, were used to investigate the degree of wear that could occur.
[31]	The results of the reaction force of the hip joint during salat were used for the simulation load parameter in the simulation of the bipolar artificial hip joint UNDIP, with those loads using the finite element method.
[32]	The effects of cochlear implants on residual hearing loss were investigated through a finite element model of human auditory periphery, consisting of the cochlea and the middle ear.
[33]	Estimation of hip implant wear, using the finite element method.
[34]	This work investigated the residual limb tension of the compression/release stabilized socket of a transfemoral amputee, using finite element modeling.

**Table 2 bioengineering-10-00268-t002:** User comfort.

Activity	Answers	No. Items	Comfort
Walk	9	12	25%
Sit down	10	12	17%
Get up	10	12	17%
Up or down	11	12	8%

**Table 3 bioengineering-10-00268-t003:** QFD Results.

Customer Requirements	Design Requirements	Concepts	Importance
Autonomy	Stability	Independence, Support, Material	1%
Safety	Resistance	Material, Simulation Test	2%
Comfort	Ergonomics	Soft material, Non-intrusive, Intuitive design	4%
Esthetics	Color	Material	5%
Usability	Shape	Function, Replicable	3%

**Table 4 bioengineering-10-00268-t004:** Material properties and simulation parameters.

	Model 1	Model 2	Model 3
**Body**	Surface thickness: 5 mm	Solid	Solid
**Material**	30% wt glass-fiber-reinforced polypropylene E= 1.3 GPa ν= 0.4 $\sigma_{y}=$ 41 MPa ρ= 950 kg/m${}^{3}$	30% wt glass-fiber-reinforced polypropylene E= 1.3 GPa ν= 0.4 $\sigma_{y}=$ 41 MPa ρ= 950 kg/m${}^{3}$	30% wt glass-fiber-reinforced polypropylene E= 1.3 GPa ν= 0.4 $\sigma_{y}=$ 41 MPa ρ= 950 kg/m${}^{3}$ Aluminum alloy E= 71 GPa ν= 0.33 $\sigma_{y}=$ 280 MPa ρ= 2770 kg/m${}^{3}$
**Element Type**	Shell 101	SOLID 187	SOLID 187
**Elements**	291,972.00	254,482.00	829,810.00
**Nodes**	293,865.00	969,298.00	3,060,273.00
**Loads**	Force 1 $F_{tot}=$ −543.72 N $F_{x}=$ −26.94 N $F_{y}=$ −542.58 N $F_{z}=$ 22.52 N Force 2 $F_{tot}=$ −127.34 N $F_{x}=$ −78.65 N $F_{y}=$ −0.744 N $F_{z}=$ 100.13 N Force 3 $F_{tot}=$ −205.68 N $F_{x}=$ −204.9 N $F_{y}=$ −9.91 N $F_{z}=$ 14.86 N	Force 1 $F_{y}=$ −543.72 N Force 2 $F_{x}=$ −205.68 N Force 3 $F_{z}=$ −127.34 N	Force 1 $F_{y}=$ −543.72 N; Force 2 $F_{x}=$ −205.68 N Force 3 $F_{z}=$ −127.34 N
**Supports**	Fixed support, two edges, to attach the prosthesis or artificial limb.	Frictionless support, six faces, to attach the belt for hip. Cylindrical support, two faces, to secure the aluminum bracket.	Frictionless support, six faces, to attach the belt for hip. Cylindrical support, two faces, to secure the aluminum bracket. Fixed support, two faces, to attach the prosthesis or artificial limb.

**Table 5 bioengineering-10-00268-t005:** Total user comfort for ergonomic socket.

Activity	Answers	No. Items	Comfort
Walk	1	12	92%
Sit down	3	12	75%
Get up	3	12	75%
Up or down	4	12	67%

**Table 6 bioengineering-10-00268-t006:** VEU consistency ranges.

Ranks	Consistencies
5.0≥x>4.5	Excellent
4.5≥x>3.5	Good
3.5≥x>3.0	Acceptable
3.0≥x>2.5	Questionable
2.5≥x>2.0	Poor
2.0≥x>0	Unacceptable

## Data Availability

The data presented in this study are available on request from the corresponding author. The data are not publicly available, due to confidentiality.

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
