# Peer review of "Development of a Hip Joint Socket by Finite-Element-Based Analysis for Mechanical Assessment"

_bioengineering, 2023, doi:10.3390/bioengineering10020268_

Round 1

Reviewer 1 Report

The paper deals with a hip joint socket design by Finite-Element assessment. It's applied to a patient with an oncological amputation. First of all, from the literature, it takes into consideration several cases  highlighting the difficulty of putting on the socker by patients.

Then, different designs based on FEM are evaluated as scientific support to the obtained results and developing a new ergonomic fit with improvement of the correct gait during the correction of postures, improving the interaction between adaptation and user and an aesthetic adaptation that satisfies the usability factor. Finaly, the physical construction of the prototype validates the results obtained.

The work is well organised; after a careful analysis of the data present in the literature, it faces a fair evaluation of what is available in the literature and of the progress in the particular sector of the hip.

The prototype of a hip joint socket, first analytically studied with the FEM, and then realized, characterizes the paper not only from an analytical theoretical point of view, but also from an experimental one.

The study and the realization from the prototype show good results for the improvement of the use of hip joint socket by the patient.

The structural aspect is well treated, while the kinematic-dynamic analysis is lacking. So, my advice is to improve this aspect in the article: you will find some basic notions on the following two that I recommend to mention in this paper:

1) Cammarata, A., Sinatra, R., Maddìo, P.D. Static condensation method for the reduced dynamic modeling of mechanisms and structures (2019) Archive of Applied Mechanics, 89 (10), pp. 2033-2051. DOI: 10.1007/s00419-019-01560-x

2) Cammarata, A., Lacagnina, M., Sinatra, R. Closed-form solutions for the inverse kinematics of the Agile Eye with constraint errors on the revolute joint axes (2016) IEEE International Conference on Intelligent Robots and Systems, 2016-November, art. no. 7759073, pp. 317-322. DOI: 10.1109/IROS.2016.7759073

Author Response

Reply to the reviewer’s comments.

We would like to thank the reviewers for their efforts and interest in our work. Your comments will improve the quality of this manuscript and give us the opportunity to clarify these comments. We will respond to all the comments raised by the reviewers.

We have carefully reviewed and attended to all comments raised by the reviewers, as listed below, highlighted in the manuscript.

REVIEWER 1

The paper deals with a hip joint socket design by Finite-Element assessment. It's applied to a patient with an oncological amputation. First of all, from the literature, it takes into consideration several cases  highlighting the difficulty of putting on the socker by patients. Then, different designs based on FEM are evaluated as scientific support to the obtained results and developing a new ergonomic fit with improvement of the correct gait during the correction of postures, improving the interaction between adaptation and user and an aesthetic adaptation that satisfies the usability factor. Finaly, the physical construction of the prototype validates the results obtained. The work is well organised; after a careful analysis of the data present in the literature, it faces a fair evaluation of what is available in the literature and of the progress in the particular sector of the hip. The prototype of a hip joint socket, first analytically studied with the FEM, and then realized, characterizes the paper not only from an analytical theoretical point of view, but also from an experimental one. The study and the realization from the prototype show good results for the improvement of the use of hip joint socket by the patient.

The structural aspect is well treated, while the kinematic-dynamic analysis is lacking. So, my advice is to improve this aspect in the article: you will find some basic notions on the following two that I recommend to mention in this paper:

1) Cammarata, A., Sinatra, R., Maddìo, P.D. Static condensation method for the reduced dynamic modeling of mechanisms and structures (2019) Archive of Applied Mechanics, 89 (10), pp. 2033-2051. DOI: 10.1007/s00419-019-01560-x

2) Cammarata, A., Lacagnina, M., Sinatra, R. Closed-form solutions for the inverse kinematics of the Agile Eye with constraint errors on the revolute joint axes (2016) IEEE International Conference on Intelligent Robots and Systems, 2016-November, art. no. 7759073, pp. 317-322. DOI: 10.1109/IROS.2016.7759073

Thank you for your advice. The references have been added according to your instructions.

Reviewer 2 Report

In this research, the critical elements for developing a socket are identified: weight distribution, empathic design, ergonomics and anthropometry, points taking account in the design. It was analysed under mechanical design criteria through Finite Element analysis due its complex geometry. In addition Finite Element Analysis have been used by to investigated anterior cruciate ligament (ACL) stress distribution under different loading conditions and evaluating biomechanical characteristics using Finite Element analysis to confirm the new abutment system’s clinical applicability. Finally, it concludes with the validation of the new adjustment that is carried out through a perception survey focused on the author’s theory of comfort, where comfort is evaluated in the different activities that the patient performs with both sockets before and after improved design. I think that the introduction of the study provide sufficient background and include all relevant references and originality & novelty.

 All the cited references are relevant to the research as shown in Table 1 and References section.

 The manuscript is structured as follows. In Section 2, the methodology or scientific procedure used in the research to develop the new ergonomic fit is presented. The main results obtained are shown, detailing various aspects such as data collection, analysis, socket design by Finite Element Method and socket manufacture. Section 3 presents the different models analyzed using the Finite Element simulation technique. Section 4 presents the validation of socket development the study of the results obtained and its discussion. The validation stage is essential to verify the increase in the comfort of the user sensation. Finally, work conclusions are presented. I think that this research design is appropriate and the methods applying to this paper are adequately described.

 But, In order to maximize the use of the results of this study, you need to present the finite element model and design factors and apply them to analysis using the existing design techniques such as design of experiments and optimal design.

 In finite element analysis, applied material properties and boundary conditions are very important, but the reason and method for setting the basic conditions for analysis are not specifically mentioned, so I think that they need to be supplemented.

 It was used in the basket and allowed it to be semi-rigid so that the patient can have a greater degree of freedom in the activities they perform daily. Polyethylene is a semi-rigid material and allowed to help in holding the basket. Furthermore, due to its high mechanical resistance and lightness, the aluminum support allowed the reinforcement of critical areas throughout the socket according to results of simulations by finite-element method. An improved socket to hip joint was developed, taking account data about conventional socket used by user probe, through using finite element method, manufacturing the prototype to evaluate it and finally validation about better performance than conventional socket was shown. Although the results are a few clearly presented and conclusions are nearly supported by the results, it is necessary to revise the results and conclusions according to the revision of the commented contents.

Author Response

Reply to the reviewer’s comments.

We would like to thank the reviewers for their efforts and interest in our work. Your comments will improve the quality of this manuscript and give us the opportunity to clarify these comments. We will respond to all the comments raised by the reviewers.

We have carefully reviewed and attended to all comments raised by the reviewers, as listed below, highlighted in the manuscript.

REVIEWER 2

In this research, the critical elements for developing a socket are identified: weight distribution, empathic design, ergonomics and anthropometry, points taking account in the design. It was analysed under mechanical design criteria through Finite Element analysis due its complex geometry. In addition Finite Element Analysis have been used by to investigated anterior cruciate ligament (ACL) stress distribution under different loading conditions and evaluating biomechanical characteristics using Finite Element analysis to confirm the new abutment system’s clinical applicability. Finally, it concludes with the validation of the new adjustment that is carried out through a perception survey focused on the author’s theory of comfort, where comfort is evaluated in the different activities that the patient performs with both sockets before and after improved design. I think that the introduction of the study provide sufficient background and include all relevant references and originality & novelty.

 All the cited references are relevant to the research as shown in Table 1 and References section.

 The manuscript is structured as follows. In Section 2, the methodology or scientific procedure used in the research to develop the new ergonomic fit is presented. The main results obtained are shown, detailing various aspects such as data collection, analysis, socket design by Finite Element Method and socket manufacture. Section 3 presents the different models analyzed using the Finite Element simulation technique. Section 4 presents the validation of socket development the study of the results obtained and its discussion. The validation stage is essential to verify the increase in the comfort of the user sensation. Finally, work conclusions are presented. I think that this research design is appropriate and the methods applying to this paper are adequately described.

  • But, In order to maximize the use of the results of this study, you need to present the finite element model and design factors and apply them to analysis using the existing design techniques such as design of experiments and optimal design.

Thank you for your feedback. It was added in Figure 3 and Table 4, with their respective explanation.

  • In finite element analysis, applied material properties and boundary conditions are very important, but the reason and method for setting the basic conditions for analysis are not specifically mentioned, so I think that they need to be supplemented.

Thank you for your comment. It was attended to in Table 4.

  • Although the results are a few clearly presented and conclusions are nearly supported by the results, it is necessary to revise the results and conclusions according to the revision of the commented contents.

We appreciate your comment, conclusions were improved.

Reviewer 3 Report

The paper is interesting and, certainly, could be useful to the community, however, in my opinion, substantial improvements need to be made before publication.

I will follow the numbering as much as possible, without distinguishing between minor and major revisions.

64-75, the citations should better relate to the research presented.

105-106, RULA and REBA should be better defined (possibly, some references can be added).

108-109, the procedure using the photographs is not clear, what method was used for the analysis and comparison of the photographs and how were the reported results obtained?

111, what does "maximum risk" mean? What kind of risk are you referring to?

114, on what basis was the score "four" obtained?

124, what is meant by "natural posture"? In other words, what is the reference?

128-129, this conclusion is also not clear.

137, "his". It's right?

150, as above.

193-195, the text speaks of a "sensor system" but does not describe how it is made, where it is positioned and, above all, what exactly does it measure? with what resolution? Are force measurements static or dynamic? How are they mediated? etc. etc.

198-199, the distribution of weights is mentioned in the text, but does not appear to be reported in the rest of the paper.

204, 208 and 212 refer to the meshes which should be illustrated in Fig.4. Unfortunately, no mesh type is visible in Fig.4. Also, the finite elements used should be better highlighted. Errors in the reported values for the number of nodes and elements are likely. Even the 3 models are not sufficiently described. It is not immediate to understand how they are worn by the patient. Why do models lack symmetry? what are the main differences between them? and, above all, with what guiding criteria these typologies were chosen. Finally, a schematic with maximum part sizes might be helpful.

200, the reference system xyz is not shown in the figures and therefore it is not possible to identify what one is looking at. This makes it difficult for this referee to continue reading the work and make further suggestions. So I keep going while I can.

Further information and clarifications should be given on the precise meaning of total deformation, structural error (which is not obvious), safety factor (it must be defined) and all the other parameters represented in the figures.

219-220, it is not clear how the loads measured with the Flexi force Pressure Sensor add up, in which positions? In the figures the symbols are too small and difficult to read. What is "fixed media"? What happened to the artificial limb? Where is it connected? Doesn't it provide a reaction force?

from 226, to which point do the displacements resulting from the model analysis refer? In the text it says that the simulated deformations are compared with those measured on the prototype, how are they measured on the prototype?

Deformations are measured in mm, but what is the need to provide 4 or even 5 digits after the decimal point? Will comparison with real measurements make sense?

The forces are also indicated in N with two digits after the decimal point (what is the measurement uncertainty of the sensor?).

The text speaks of material failure, but no failure criterion is indicated.

Comparisons between displacements are made in absolute terms, wouldn't it be appropriate to speak in terms of strain and make comparisons in percentage terms?

Fatigue damage is mentioned. But it doesn't seem to me that the fatigue tests are described.

Has creep been considered?

Also, what material is the screw made of? Has relaxation been considered? It would be appropriate to describe these details well. From what I understand, I don't see the need to report all the images of the various attempts made on the holes in figure 8 (what do they add? since their positioning is arbitrary). It would be useful to have indications regarding a methodology for optimizing the position, size and number of holes.

In the experimental part it would be useful, as well as clarifying, to show (for a visual comparison) the ergonomic and the conventional prototype together.

Author Response

Reply to the reviewer’s comments.

We would like to thank the reviewers for their efforts and interest in our work. Your comments will improve the quality of this manuscript and give us the opportunity to clarify these comments. We will respond to all the comments raised by the reviewers.

We have carefully reviewed and attended to all comments raised by the reviewers, as listed below, highlighted in the manuscript.

REVIEWER 3

  • The work describes a design methodology of the hip joint socket using finite element analysis. The article is interesting and certainly, could be useful to the scientific community. However, in my opinion, substantial improvements need to be made before publication.

The observation is appreciated, and the improvements made were: 

  1. a) 3 more bibliographical references were added.
  2. b) The design methodology was depicted.
  3. c) Material properties, boundary conditions, and other considerations for FEM were included in a new Table.
  4. d) The use of FEM or FEA according to the context.

e) Conclusions were improved.

Reviewer 4 Report

Authors presented prosthesis socket design and analysis using FEA. This reviewer feels that the design optimization process should be formulated more rigorously in a mathematical form. In addition, the following improvements are suggested. 

1. Page 2 lines 66 and 67, Finite Element Analysis/Method were used several times in the paper, their abbreviation FEA or FEM should be used instead because these abbreviations are commonly used in the literature. 

2. Page 3 line 109, why 200 photos were taken? This can be done easily by using motion captures. 

3. Page 4 Figure 2, for knee flexions, the conventional and economic designs are almost identical, why?

4. Page 7 Figure 4, the mesh of the model cannot be seen from the figure. It is solid model and there are no meshes. 

5. Page 6 section 2.3, for the socket design problem, please formulate this design problem as a standard math optimization problem with design variables, objective function and constraints. 

6. After formulating the optimization problem, analyze the designs based on the optimization results. In addition, please provide boundary conditions, external loads, and optimization solver and performance. 

Author Response

Reply to the reviewer’s comments.

We would like to thank the reviewers for their efforts and interest in our work. Your comments will improve the quality of this manuscript and give us the opportunity to clarify these comments. We will respond to all the comments raised by the reviewers.

We have carefully reviewed and attended to all comments raised by the reviewers, as listed below, highlighted in the manuscript.

REVIEWER 4

Authors presented prosthesis socket design and analysis using FEA. This reviewer feels that the design optimization process should be formulated more rigorously in a mathematical form. In addition, the following improvements are suggested. 

  • Page 2 lines 66 and 67, Finite Element Analysis/Method were used several times in the paper, their abbreviation FEA or FEM should be used instead because these abbreviations are commonly used in the literature.

The authors appreciate the comment thus, the abbreviation FEA or FEM was utilized in the paper.

  • Page 3 line 109, why 200 photos were taken? This can be done easily by using motion captures.

Thank you for this observation. The comment is true. We made our best to accomplish the tests with the equipment and infrastructure of our institution. 

  • Page 4 Figure 2, for knee flexions, the conventional and economic designs are almost identical, why?

The authors appreciate the comment. The reason is that the user experience methodology remarks such as conventional flexions. This methodology has threatened the optimization of the mechanical issues considering the comfort of the user to improve the load cycle in the socket.

  • Page 7 Figure 4, the mesh of the model cannot be seen from the figure. It is solid model and there are no meshes.

The authors appreciate the comment. The figure was changed to show the mesh.

  • Page 6 section 2.3, for the socket design problem, please formulate this design problem as a standard math optimization problem with design variables, objective function and constraints.

We appreciate the observation made, and It was added in Figure 3, with its respective explanation.

  • After formulating the optimization problem, analyze the designs based on the optimization results. In addition, please provide boundary conditions, external loads, and optimization solver and performance.

Thank you for your appreciative feedback. Your comment was taken into account to complement the work added the table 4.

Round 2

Reviewer 3 Report

I find the new version satisfactory,

good luck.